# 'We are all serving the same Ugandans': A nationwide mixed-methods evaluation of private sector surgical capacity in Uganda

Katherine Albutt[1,2]*, Gustaf Drevin[2,3], Rachel R. Yorlets[2,4], Emma Svensson[2,5], Didacus B. Namanya[6,7], Mark G. Shrime[2,8], Peter Kayima[9,10]

**1** Department of Surgery, Massachusetts General Hospital (MGH), Boston, MA, United States of America, **2** Program in Global Surgery and Social Change (PGSSC), Harvard Medical School, Boston, MA, United States of America, **3** Department of Public Health Sciences, Karolinska Institutet, Solna, Sweden, **4** Department of Plastic and Oral Surgery, Boston Children's Hospital, Boston, MA, United States of America, **5** Department of Clinical Sciences, Faculty of Medicine, Lund University, Lund, Sweden, **6** Ministry of Health (MOH), Kampala, Uganda, **7** Uganda Martyrs University (UMU), Nkozi, Uganda, **8** Center for Global Surgery Evaluation, Massachusetts Eye and Ear Infirmary, Boston, MA, United States of America, **9** Mbarara University of Science and Technology (MUST), Mbarara, Uganda, **10** St. Mary's Lacor Hospital, Gulu, Uganda

* kalbutt@partners.org

**Data Availability Statement:** All relevant data are within the manuscript and its Supporting Information files.

## Abstract

### Introduction

Half of all Ugandans (49%) turn to the private or private-not-for-profit (PNFP) sectors when faced with illness, yet little is known about the capacity of these sectors to deliver surgical services. We partnered with the Ministry of Health to conduct a nationwide mixed-methods evaluation of private and PNFP surgical capacity in Uganda.

### Methods

A standardized validated facility assessment tool was utilized to assess facility infrastructure, service delivery, workforce, information management, and financing at a randomized nationally representative sample of 16 private and PNFP hospitals. Semi-structured interviews were conducted to qualitatively explore facilitating factors and barriers to surgical, obstetric and anaesthesia (SOA) care. Hospitals walk-throughs and retrospective reviews of operative logbooks were completed.

### Results

Hospitals had a median of 177 beds and two operating rooms. Ten hospitals (62.5%) were able to perform all Bellwether procedures (cesarean section, laparotomy and open fracture treatment). Thirty-day surgical volume averaged 102 cases per facility. While most hospitals had electricity, oxygen, running water, and necessary equipment, many reported pervasive shortages of blood, surgical consumables, and anesthetic drugs. Several themes emerged from the qualitative analysis: (1) geographic distance and limited transportation options delay reaching care; (2) workforce shortages impede the delivery of surgical care; (3)

**Funding:** This work was supported by: MGH Global Surgery Fund (KA); MGH Center for Global Health (KA); Ronda Stryker and William Johnston Global Surgery Fellowship (KA); Program in Global Surgery and Social Change (KA, ES, GD); Swedish Society of Medicine (GD); and Erik and Göran Ennerfelt, Fredrik Lindström, and Carl Erik Levin foundations (GD).

**Competing interests:** The authors have declared that no competing interests exist.

emergency and obstetric volume overwhelm the surgical system; (4) medical and non-medical costs delay seeking, reaching, and receiving care; and (5) there is poor coordination of care with insufficient support systems.

## Conclusion

As in Uganda's public sector, barriers to surgery in private and PNFP hospitals in Uganda are cross-cutting and closely tied to resource availability. Critical policy and programmatic developments are essential to build and strengthen Ugandan surgical capacity across all sectors.

## Introduction

Conditions requiring surgical, obstetric, and anaesthesia (SOA) services amount to a third of the global disease burden, yet over two-thirds of the world's population lack access to safe, timely, and affordable SOA care when needed [1]. Research and advocacy over the past several years have highlighted the scope and seriousness of the surgical disease burden in low- and middle-income countries (LMICs) [1–3].

Despite significant strides, multiple barriers still prevent the provision of safe, affordable, and timely surgery to those who need it. Globally, the poorest one-third of the world, where most of the surgical disease burden resides, receives only 6% of surgical procedures worldwide [1]. This is especially true in Uganda, where existing data suggests access to surgical services is severely limited, largely attributable to constraints in infrastructure, service delivery, and workforce [4–8].

We partnered with the Ministry of Health (MOH) in Uganda in 2016 to better understand the capacity of the public healthcare system to deliver safe, timely, and affordable surgical care [5, 6]. Results from this study highlighted significant delays in accessing surgical care, critical workforce shortages, inadequate infrastructure, overwhelming emergency and obstetric volume, supply chain inefficiencies and catastrophic expenditures for patients and their families. However, half of all Ugandans (49%) utilize private or private-not-for-profit (PNFP) sector care but little is known about the capacity of these sectors to deliver surgical services [8]. A review of available surgical assessments in sub-Saharan Africa reveals that only 29.8% of facilities assessed were private or PNFP despite the fact that these sectors account for as much as half of health care provision on the continent, and their role is growing [4, 9–38].

Recognizing this gap in knowledge, we partnered with the MOH to conduct a nationwide stratified randomized mixed-methods evaluation of private and PNFP surgical capacity and barriers to the provision of surgical care in Uganda.

## Methods

The data presented in this manuscript are derived from a mixed-methods nationwide study conducted in Uganda from July to October 2017.

### Data collection

A total of 16 private and PNFP hospitals in Uganda were selected using stratified purposive sampling. At the time of data collection, there were 59 public hospitals and 90 private and PNFP hospitals in Uganda [8]. In order to capture a representative sample of hospitals, two

smaller hospitals (total inpatient beds less than 200) and two larger hospitals (total inpatient beds greater than or equal to 200) within each of Uganda's four regions were randomly chosen for data collection (the exception to this rule was in the Eastern region where Kamuli only has 160 beds but is the second largest private/PNFP in the region). All assessments took place at the interviewee's hospital and were conducted by a study team member (KA) with assistance from multinational collaborators (PK, DN, and GD). Site visits were conducted at one facility per day and averaged 3–6 hours.

## Definitions

A surgical procedure was defined as "the incision, excision, or manipulation of tissue that needs regional or general anaesthesia, or profound sedation to control pain" [1]. A Bellwether procedure was defined as cesarean delivery, laparotomy, or open fracture treatment; procedures that serve as a proxy for surgical systems that are functioning at a level of complexity advanced enough to do most other surgical procedures [1]. Access to necessary inputs in the surgical system was defined as always (100%), almost always (76–99%), most of the time (51–75%), sometimes (26–50%), rarely (1–25%), and never (0%).

## Quantitative study procedures and analysis

We administered the Surgical Assessment Tool (SAT), a standardized validated facility assessment survey, at each hospital to assess facility infrastructure, service delivery, workforce, information management, and financing systems [39, 40]. The SAT was piloted at Mbarara Regional Referral Hospital in western Uganda to ensure context-specific applicability and subsequently utilized in a 2016 nationwide surgical assessment of the public sector [5, 41]. Information was collected through a combination of direct observation during hospital walk-throughs, as well as through in-person interviews., All operative cases recorded in the 30 days prior to the site visit were de-identified and manually coded in a comprehensive database.

Data were collected and managed with KoboToolbox [42]. R Computing Language and Stata 12.1 were used for data analysis (R Core Team, Vienna, Austria and Stata Corp, College Station, TX, USA, respectively). Fisher's exact and two-sample t-tests were used for comparisons.

## Qualitative study procedures and analysis

The qualitative portion of the assessment was comprised of semi-structured interviews regarding infrastructure, service delivery, surgical workforce, information management, and financing of surgical care [S1 Appendix]. During each hospital site visit, all available hospital directors, administrators, surgeons, obstetricians, anesthesiologists, and principal nursing officers were invited to participate in a qualitative interview. The most senior-level provider, or their designee, was interviewed within each cadre; none declined participation. Verbal informed consent was obtained prior to enrollment. Interviews were conducted in person in English (an official language in Uganda) in private and responses were recorded and subsequently transcribed. Duration ranged from 30 to 60 minutes. Interview transcriptions were cross-checked for accuracy and completeness by a second member of the study team.

Twenty-two interviews were completed with 27 participants at 16 hospitals (Fig 1). Participants included nine hospital administrators or medical directors, 12 physicians (surgeons, anesthesiologists, or obstetrician/gynecologists), and six other designees (nurses, anaesthesia officers, finance officers, and laboratory personnel). Of these interviews, 17 were performed at PNFPs versus five at private facilities and 11 each at larger and small hospitals, respectively. For logistical reasons, three interviews were conducted in groups that involved hospital

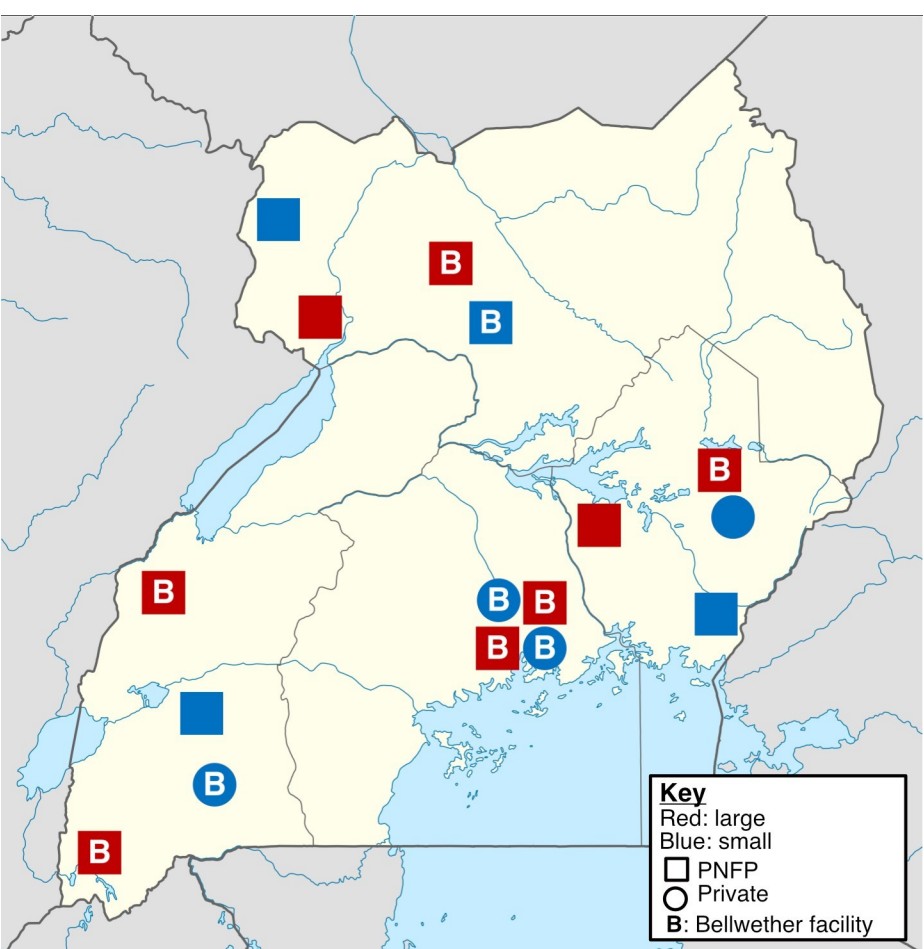

**Fig 1. Map of sampled facilities in Uganda.**

administrators, medical directors, physicians, nurses, and others. Group interviews were moderated to facilitate input from all participants and across all cadres. Care was taken to seek input across multiple cadres to avoid the limitations associated with a single perspective. Interview recordings were transcribed verbatim and uploaded to the qualitative data analysis software (QSR International Nvivo 11) [43]. Hospital metrics are presented in Table 1.

Using grounded theory, a comprehensive codebook was created independently by two authors (RY and ES). This manual was then used by the primary analysts (RY and ES) to code all subsequent transcripts with independent validation conducted by a third analyst (KA). Thematic content analysis was used as the analytic approach and saturation was achieved. Coding inter-rater reliability, measured with a pooled Cohen's kappa, was 0.83 [44].

### Ethical considerations

This study was deemed exempt from review by the Institutional Review Board (IRB) at Boston Children's Hospital and approved by the IRBs at Mbarara University of Science and Technology and the Uganda National Council for Science and Technology. Additional approval was obtained from the Ministry of Health and Uganda Catholic Medical Bureau, where appropriate.

**Table 1. Hospital metrics.**

| Hospital | Sector | Region | Beds | Operating theatres | ICU beds | One month operative volume |
|---|---|---|---|---|---|---|
| Galilee Community General Hospital | Private | Central | 18 | 1 | 0 | 7 |
| St. Francis Hospital Nsambya | PNFP | Central | 361 | 1 | 16 | 343 |
| Paragon Hospital | Private | Central | 29 | 1 | 0 | 29 |
| Lubaga Hospital | PNFP | Central | 240 | 4 | 0 | 228 |
| Pope John XIII Aber Hospital | PNFP | North | 178 | 1 | 0 | 67 |
| St. Mary's Lacor Hospital | PNFP | North | 482 | 6 | 10 | 425 |
| St. Luke's Hospital | PNFP | North | 220 | 2 | 0 | 148 |
| Oriajini Hospital | PNFP | North | 60 | 1 | 0 | 1 |
| Ibanda Hospital | PNFP | Western | 176 | 2 | 0 | 82 |
| Mutolere Hospital | PNFP | Western | 210 | 2 | 0 | 208 |
| Mayanja Memorial Hospital | Private | Western | 100 | 2 | 0 | 97 |
| Virika Hospital | PNFP | Western | 205 | 2 | 4 | 106 |
| Kanginima Hospital | Private | Eastern | 25 | 4 | 0 | 11 |
| Kumi Hospital | PNFP | Eastern | 300 | 4 | 0 | 166 |
| Kamuli General Hospital | PNFP | Eastern | 160 | 3 | 0 | 136 |
| Dabani Hospital | PNFP | Eastern | 80 | 1 | 0 | 45 |

# Results

## Access to care

Hospital catchment populations were described as large (as many as several million people), far-reaching, and occasionally inclusive of refugee camps and neighboring countries. Only six facilities (37.5%) reported that patients were almost always able to access the hospital within two hours. Providers consistently identified long distances and limited transportation as barriers to reaching care. They note that patients are often forced to use or borrow personal vehicles, walk, or transit on motorcycles (commonly known as *boda boda*), delaying presentation. As one provider described, *"They* [patients] *came in late and during the resuscitation they passed on."* Another said, *"Patients have been dying before surgery. For example, somebody has come with bleeding maybe from blunt abdominal trauma, the ruptured uterus, and they came late. They are very pale, they're in shock. And before we even do assessment—dead."* Delays arise while seeking, reaching, and receiving appropriate care.

## Infrastructure

Median bed capacity was 177 beds (IQR 75–225). PNFP hospitals had a greater capacity with a median of 207.5 (IQR 272–255) beds versus 27.0 (IQR 23.25–46.75) beds in the private sector. There was a median of 2.0 (IQR 1–3.25) functional operating rooms (ORs) per facility. Most facilities (56.3%) lacked a dedicated area for post-anaesthesia care; instead, patients recovered in the ward, theater, or adjacent areas, monitored by theater nurses. Three hospitals had a functional intensive care unit (ICU), ranging from four to 16 beds. Many providers expressed the need and wish to develop an ICU or high-dependency unit.

Most hospitals reported access to electricity (87.5%), oxygen (93.8%), and running water (93.8%) at least half of the time(Fig 2). Electricity is usually sourced from the national grid, although many hospitals are supported by generators during frequent power outages. A total of 5 hospitals (31.3%) lacked electricity more than 25% of the time. Facilities source running water through the national water supply, boreholes (drilling), or rain harvesting; but one facility rarely had running water (less than 25% of the time). Oxygen supply across facilities was

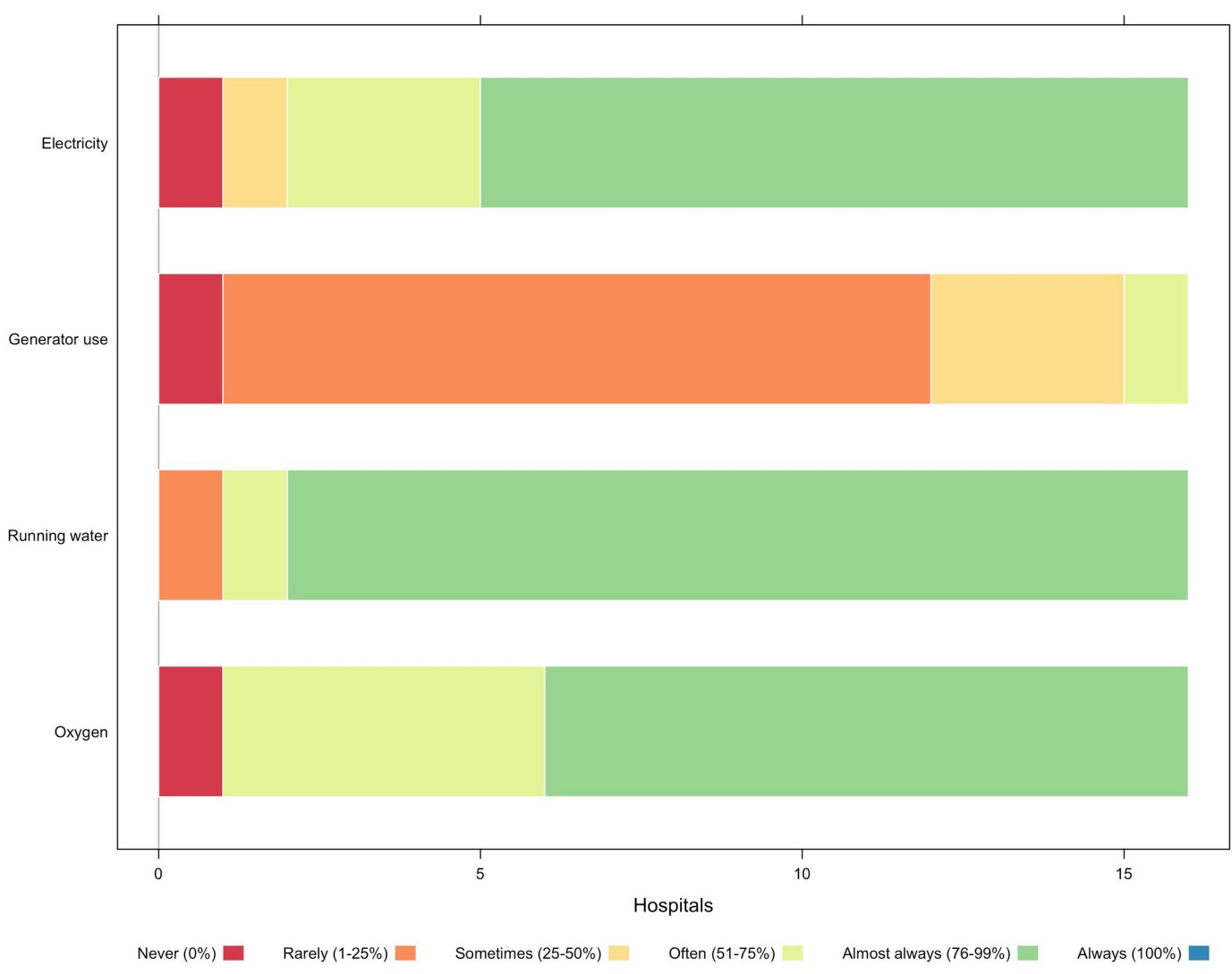

**Fig 2. Availability of surgical infrastructure.**

less consistent, largely reflective of supply chain issues. A total of 6 hospitals (37.5%) lacked oxygen more than 25% of the time, including one that never had access to oxygen.

## Supply chain

Private and PNFP facilities rely on Joint Medical Stores (leading private pharmaceutical store in Uganda serving at least 3000 medical facilities; not-for-profit joint venture between Ugandan Catholic Medical Bureau and Uganda Protestant Medical Bureau), donations, and the open market for supplies. Most hospitals reported a steady supply of medications and consumables but noted that stock-outs of essentials, such as anesthetic and analgesic drugs, alcohol scrub, and chest tubes, occur (Fig 3). Some facilities identified workarounds to supply chain shortages, by pre-qualifying a list of back-up drug and medical suppliers or improvising (e.g., using nasogastric tubes instead of chest tubes). While donations might provide equipment not otherwise available, the supply is unpredictable and unresponsive to need.

## Equipment

Most hospitals reported having necessary equipment, such as ventilators, pulse oximetry, suction machines, electrocautery, and autoclaves (Fig 4). The median number of functioning ventilators per facility was 2.0 (IQR 1–2). Only half of hospitals had 24-hour access to radiology services. In most facilities, ultrasounds and X-rays were the only radiographic services available. A CT scanner was available in only 18.7% of sampled hospitals, forcing providers to refer patients for CT scans. A number of participants mention more comprehensive radiology services as an outstanding need, and many expressed a wish to procure a CT scan for the hospital.

## Blood

Pervasive blood shortages at the regional blood banks and lack of blood banks at any of the hospitals often prevent timely blood transfusions:

> *You go to the regional bank—blood is not there. But sometimes you get at least an adequate amount, though it's suffering. When the students are in schools, that's when we have blood. When they are on holiday, there is no blood.*

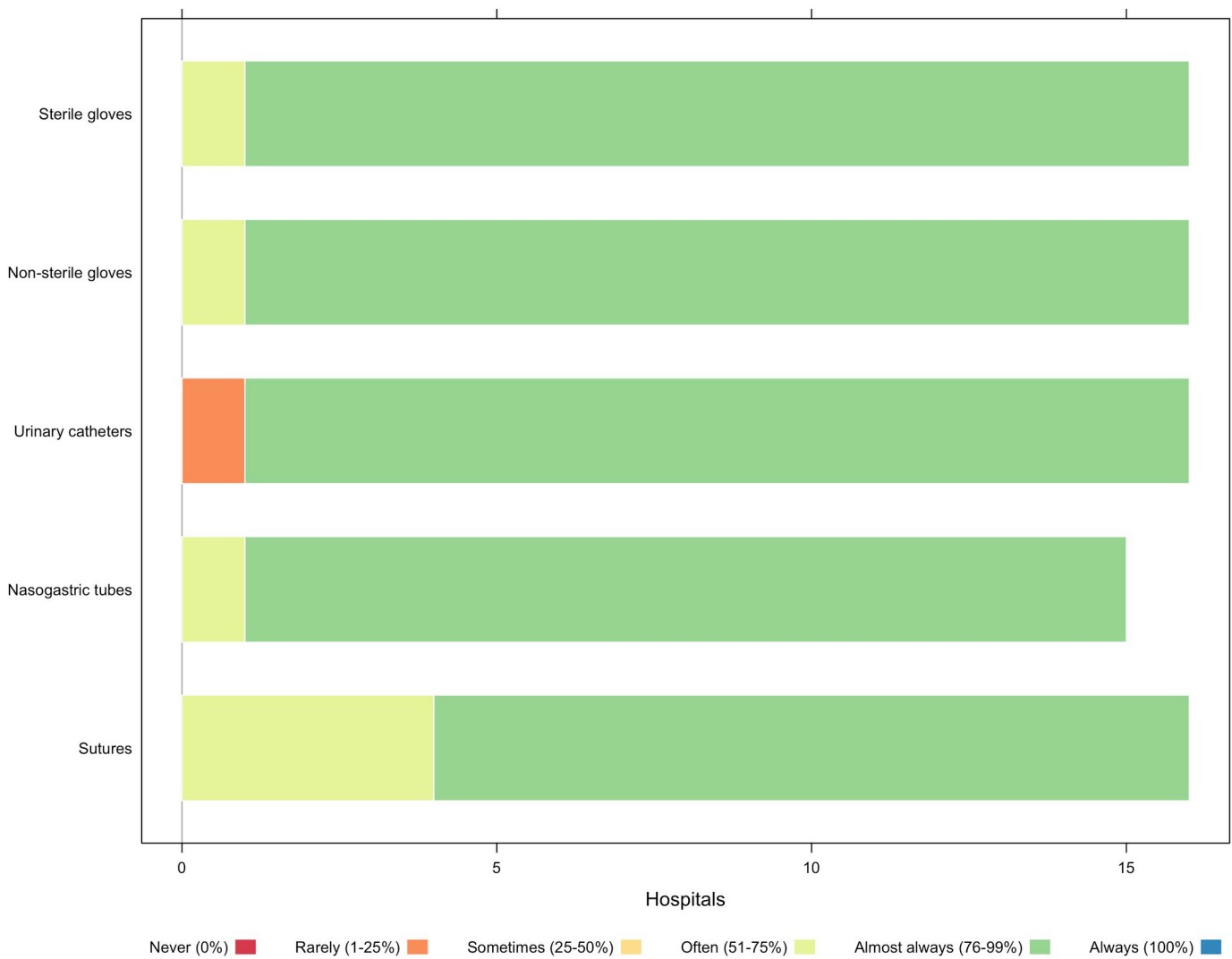

**Fig 3. Availability of consumables.**

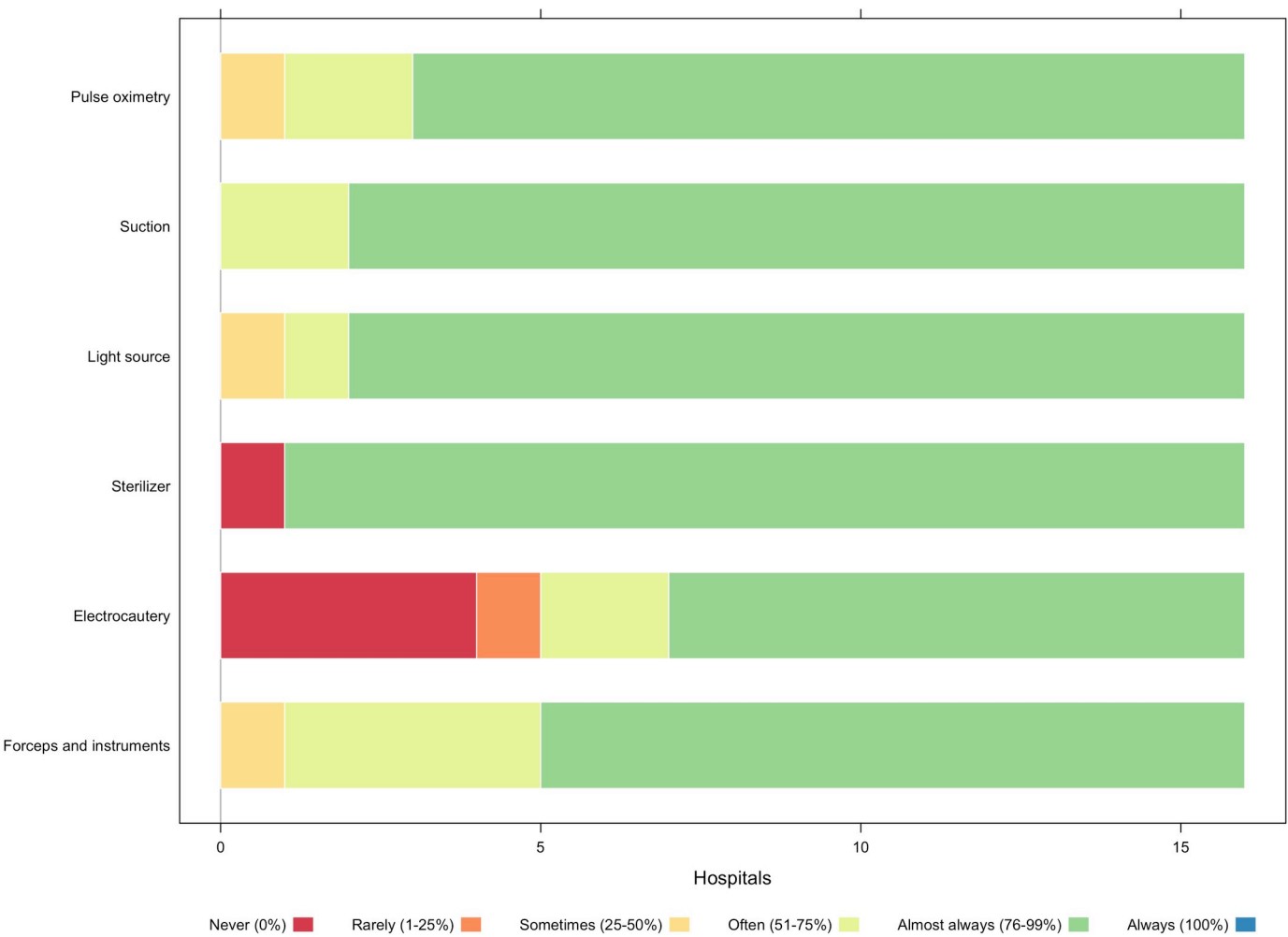

**Fig 4. Availability of surgical equipment.**

Blood shortages often postpone or prevent surgical care, sometimes resulting in death:

*It's a big challenge . . .sometimes you have to postpone operations or actually cancel operations which may be needed, whose transfusion requirements may be high. We have also lost patients sometimes because there was no blood transfusion given. It's usually mothers coming who have had ruptured uterus or have had postpartum hemorrhage. We've lost people because there is no blood.*

## The surgical workforce

There were 138 qualified SOA physicians employed at sampled hospitals (median 7, IQR 6–12.5) at least one day per week. Even when employed, SOAs were frequently unavailable around the clock. Medical officers (non-specialized physicians who have completed internship) and anaesthetic officers (anaesthesia providers without a medical degree who have completed a diploma) were available 24 hours a day in 12 (75.0%) and 13 hospitals (81.3%), respectively. In nearly all hospitals (93.8%), medical officers performed surgeries.

The surgical workforce is understaffed, particularly in terms of SOA specialists. Access to providers was limited with only eight hospitals (50%) reporting access to a surgeon, eight (50%) reporting access to an obstetrician and four (25%) reporting access to an anesthesiologist at least half the time. One provider illustrated how this affects quality of care:

*We are not well-staffed. . . .we are supposed to be having ten doctors ideally here, but we have only two medical officers and me. Having to look after all these patients, it's a big challenge, it's an uphill task. Sometimes it's also a big problem for me. . . I think there is a lot of pressure. It's a lot of pressure on me; it's overwhelming. So, sometimes, I may not be available to help each and everybody as it should be.*

To address these issues, facilities partner with external providers and/or facilities to create a 'stand-by' group of surgical or anesthetic providers that they can call when needed. This often creates staffing challenges for both public and private facilities as providers are pulled between both.

In PNFP hospitals, constrained budgets often prevent recruitment and retention of an adequate surgical workforce. Low pay and long hours affect morale, as one provider illustrated: *"You find that you are just working for God and really, for you, you can't even [pay fees in order to] take your child to a good school."* Many providers mentioned that they would like to invest in more continuing education for staff to improve the quality of care and improve retention and satisfaction. Others have turned to training internal staff as a means to retain their workforce, building in "bonding agreements" to their training contracts.

## Service delivery

The private and PNFP sectors offers a wide range of SOA services. Operative volume is presented in Table 2. A total of 2,099 procedures were recorded across sites in the 30-day period prior to each site visit, a median 102 (IQR 41–176.5) procedures per facility. A total of 6.9% of the procedures (n = 144) were recorded at private facilities and 93.1% (n = 1955) at PNFP facilities. Only ten hospitals (62.5%) were capable of performing all three Bellwether procedures. The total case mix was predominantly obstetrics and gynaecology (61.0%; n = 1281), followed by general surgery and trauma (31.2%; n = 655), followed by orthopedic (7.8%; n = 163). Cesarean deliveries are the most common procedures at most hospitals (46.6%; n = 979). While laparotomies (7.3%; n = 153) and fracture repairs (2.4%; n = 51) are performed, albeit much less frequently, they are often performed by the external stand-by specialists. Providers note that this is particularly common for orthopedic procedures: patients are often temporized and referred, asked to wait for an orthopedic specialist to be called in from another hospital, or are treated by a non-specialist.

Up to 90% of surgeries are emergencies. These include orthopedic and neurosurgical treatment of traumatic injuries, caesarean sections, and exploratory laparotomies for typhoid perforations, among others. Participants often referenced the burden of trauma: *"We really need to support at least the injured, the trauma patients. Yes, because we have talked about tuberculosis, we have talked about malaria. . .but the biggest killer, we are not talking about it. In fact, trauma should be a priority."* Several hospitals also perform elective surgery including cancer, thyroid, and gynecologic operations. Caseload varies by season and facility accessibility. Furthermore, the caseload in the private and PNFP sectors is affected by referrals from public hospitals:

*And in terms of burden, sometimes we feel lot of pressure. Sometimes, government hospitals, I'm sorry to say this, are not functional. So, you find their, all their ambulances coming here*

**Table 2. Operative volume.**

| Procedure | Private One Month Operative Volume | PNFP One Month Operative Volume | Total One Month Operative Volume |
|---|---|---|---|
| Caesarean section | 109 | 870 | 979 |
| Hysterectomy | 0 | 65 | 65 |
| Exploratory laparotomy | 4 | 46 | 50 |
| Salpingectomy/salpingoophorectomy | 0 | 8 | 8 |
| Tubal ligation | 0 | 2 | 2 |
| Other | 7 | 170 | 177 |
| *Total obstetrics and gynaecology* | *120* | *1161* | *1281* |
| Skin and soft-tissue excision or biopsy | 2 | 104 | 106 |
| Exploratory laparotomy | 4 | 99 | 103 |
| Incision and draiange and/or debridement | 4 | 96 | 100 |
| Hernia repair | 8 | 90 | 98 |
| Hydrocelectomy | 0 | 16 | 16 |
| Haemorrhoidectomy | 0 | 12 | 12 |
| Skin graft | 0 | 13 | 13 |
| Mastectomy or lumpectomy | 0 | 8 | 8 |
| Burr hole | 1 | 5 | 6 |
| Other | 3 | 190 | 193 |
| *Total general surgery and trauma* | *22* | *633* | *655* |
| Operative fracture repair (including exfix) | 1 | 50 | 51 |
| Amputation | 0 | 21 | 21 |
| Incision and draiange and/or debridement | 0 | 11 | 11 |
| Casting or splinting | 1 | 12 | 13 |
| Other | 0 | 67 | 67 |
| *Total orthopaedic surgery* | *2* | *161* | *163* |
| **Total** | **144** | **1955** | **2099** |

*to bring patients for caesarean section, for intestinal obstruction . . . And the reason they high-light: 'you see, we don't have this, or the doctor is not there, we don't have supplies, the anaes-thetist is not available at station.' But we have always attended to these patients.*

## Cost of care

Providers note that the inability to afford care influences patient's care-seeking behavior, often causing patients to delay seeking care. Some hospitals openly acknowledge these financial bar-riers and try to overcome them by providing care for free or at lower, subsidized rates.

Hospitals still treated patients who were unable to pay, especially if it was an emergency. One participant explained, *"If somebody came in and really needs the caesarean section and cannot afford the bill, and you really talk to them and they cannot afford, sometimes we intend to try to save life and we think about other things later."* Another remarked, *"Being on the road-side, we receive accident cases. Someone is a good Samaritan and picks somebody who has had an accident. We treat and in the end the patient does not have relatives, no attendants. Who is going to cater for the bill? No one."*

There was no clear consensus on how patients who cannot pay should be handled. While some hospitals referred these patients, others did not charge them, established payment plans, covered the direct medical costs through a compassionate fund, or detained patients until they or their families can procure payment.

*The first thing which we do is to refer the patient to the public hospital. But most of the patients have been referred from the public hospital to come here. And sometimes, if it's an emergency you can't refer back. You have to continue and work on. If they know they are unable to pay, then you request them to try and pay something. And in the end, if they are still here and they can't pay, the administration looks into it, and they release them.*

Another said:

*So, we offer treatment but then they escape, some fail to pay... and people run away. The attendant, the relatives, they run away and go... so, we face those challenges. That is the problem we have. At long last we release the patients. What can we do? ...But surprisingly they just discharge themselves. It is a big challenge.*

Numerous participants described their institution's religious mission to care for all patients, regardless of ability to pay. In the PNFP sector, many facilities are governed by either the Uganda Catholic Medical bureau or the Uganda Protestant Medical Bureau. One surgeon explained, *"Being a church-based institution, everyone is equal. You can't say you are not going to treat me because I don't have any money. We treat everyone. It's a challenge."*

### Referral system

Inadequate coordination of care, including a malfunctioning referral system, exacerbates the aforementioned problems. Participants explained that patients are sometimes referred from higher-level facilities that lack the capacity to treat a patient, without inter-institution communication, often because a facility lacks the necessary workforce, supplies, or a specific diagnostic test. Referral patterns often depend on the presence or absence of specialist providers. One participant stated, *"Yesterday when I was on call, I received a mother who was obstructed and somebody gave medication and referred, without even escorting. Such cases are very common. They don't communicate. Just, just leave them here. ...The referral system is not working."*

Participants consistently noted the lack of accessible patient transportation and requested MOH support in terms of policy and programming to address this unmet need. Deficient ambulance fleets limit the transportation of patients to or between hospitals. Participants also saw other roles for the MOH to oversee and facilitate coordination of care via financial support, workforce expansion, and improved trauma care, amongst others. One participant said, *"You know, we are all serving the same Ugandans. I think the government is trying but we should get a bit more prerogative."*

## Discussion

This nationwide assessment of surgical capacity in Uganda's private and PNFP sectors revealed common barriers to delivering surgical care, with some variation facility to facility. Major challenges can be grouped broadly into five themes: (1) geographic distance and limited transportation options delay reaching care, (2) workforce shortages impede surgical service delivery, (3) emergency and obstetric volume overwhelm existing capacity, (4) medical and non-medical costs delay seeking, reaching, and receiving care, and (5) there is poor coordination of care with insufficient support systems.

### Geographic distance and limited transportation options delay reaching care

Barriers to seeking, reaching, and receiving surgical care have been qualitatively assessed across numerous low-resource settings, including Uganda's public sector [6, 45]. Poor road

infrastructure and limited transportation options prolong transit and referral times to facilities that perform surgery [46]. Surgically-capable hospitals are scarce and may be hours or days away. Clinicians and administrators in Uganda consistently cited geography and limited transportation as critical access barriers, noting that delayed presentation results in more advanced pathology and worse outcomes. This is consistent with Uganda's public sector as well as other low-resource settings around the globe [6, 45, 46].

## Workforce shortages impede surgical service delivery

As in other low-income settings, Uganda faces a surgical workforce crisis. As of 2015, less than one-fifth of the surgical workforce in LMICs (19% of surgeons, 15% of anaesthesiologists, and 29% of obstetricians) are responsible for meeting the needs of half of the world's population [47]. Uganda's private and PNFP hospitals are understaffed and the available staff are not distributed to meet the demand. Despite the fact that most Ugandans are rural, over 90% of Uganda's physicians are concentrated in the capital Kampala, creating an access chasm across the country [48].

Many private and PNFP facilities rely on external SOA providers to overcome workforce shortages–a workaround that seems to be more pervasive in Uganda's private and PNFP sectors than in other low-resource settings, including the public sector [6]. Because full-time specialists are often cost-prohibitive, most private facilities contract with per diem providers to be called as needed when surgical and obstetric patients present. However, this dependence on an external workforce delays care (specialists are not always available when called), destabilizes the public sector workforce, and complicates care continuity [6].

## Emergency and obstetric volume overwhelm existing capacity

High emergency and obstetric surgical volumes overwhelm the already limited operative capacity in Uganda's private and PNFP hospitals. An effective surgical system sustains a balance of emergency and elective surgery [49, 50]. However, in some of the assessed facilities, nearly 90% of cases are emergencies, with caesarean deliveries as the most common operation. Attending to the disproportionate volume of emergency and obstetric surgery leaves few resources to meet remaining surgical need. Because surgical care is not available in a timely manner, easily treatable conditions progress to complicated disease with high case fatality and morbidity [51].

## Medical and non-medical costs of care delay seeking, reaching, and receiving care

Globally, 81 million people face financial catastrophe every year due to the medical and non-medical costs of seeking and receiving surgical care [52]. In Uganda's private and PNFP sectors, out-of-pocket spending is the principal payment method. While care of the poorest may be subsidized for certain services, financial risk prevents and delays many from seeking, reaching, and receiving surgical care. Furthermore, variable out-of-pocket payments can be devastating for patients and their families [53, 54]. Unique to the PNFP sector are financing strategies such as compassionate funds and community payment plans which may defray a fraction of costs for those less fortunate.

## Poor coordination of care with insufficient supporting systems

Insufficient supporting systems and poor coordination of care within the system as a whole exacerbate many of the aforementioned problems. Respondents identified a clear opportunity for the MOH to support, oversee, and facilitate coordination of care on a systems level. Despite

each tending to the needs of approximately half of the population, there is no coordinated interface between or within the public, private, and PNFP sectors. Patients are referred without inter-institution communication and without knowledge of the referral facility's capacity to receive the patient. There is no guarantee that the receiving facility can provide the requisite care and no ambulance availability to facilitate transfer to or between hospitals. Further, an effective and timely surgical referral system requires a level of financing and coordination of resources that is often difficult to attain in low-income settings such as Uganda [55]. Unfortunately, the lack of coordination of care often results in delays which compound one another, ultimately resulting in worse outcomes for patients. Coordination of care between sectors has been identified as a strategic priority for the MOH going forward.

## Differences between the private and PNFP sectors and the public sector

Barriers to facility availability, geographic access, poor inter-facility coordination, overwhelming patient volumes, disproportionate emergency surgery volumes, and deficient infrastructure and consumable supply lines are common to both the public and private and PNFP sectors.[5, 6] All facilities rely on a common supplier, the same blood banks for transfusions, and the national grid for electricity. Surgical facilities in Uganda see a disproportionate volume of caesarean deliveries compared to other operative interventions, in both the public sector and private and PNFP sectors. There is no formal referral system between public, private, and PNFP facilities, and what little communication or coordination exists is informal and built on personal connections.

This study identifies several strengths of the private and PNFP sectors. Infrastructurally, private and PNFP facilities are smaller and generally better equipped. Private and PNFP facilities did not report seeing "floor cases everywhere", which is the reality in many public facilities. They seldom run out of consumables, in stark contrast to most public facilities reporting that consumables are available only "sometimes" or "rarely".[5] Public facilities have been described as older[6], larger, and not well-equipped or staffed–yet we found that 30-day as well as median facility operative volume was 50% higher in public than private and PNFP hospitals (total 30-day volume: 3014 versus 2107, respectively; median facility 30-day volume: 164 versus 102, respectively). Private and PNFP staffing ratios were higher than the public (138 versus 83 SAO specialists, respectively), mainly because of part-time locums of specialists in private and PNFP facilities and possibly higher reliance on medical officers in the public sector.[5] Finally, private facilities, which require insurance or out-of-pocket payments, offer flexibility through "compassionate funds" and community payment plans. These differences in workforce, case mix, and financial capacity highlight a need for sector-specific capacity building interventions as existing needs in public and private settings are markedly different. Potential exists to better harness the strengths of the private and PNFP sector to advance the provision of SOA care in Uganda country-wide.

## Limitations

This study has several limitations. Due to logistical and financial limitations, we were not able to assess all private and PNFP hospitals in the country, though our sample is likely representative. Additionally, collection of data with the SAT is subject to observer bias, which may have under- or overestimated the availability and access to services, equipment, or supplies in question. Interviewer bias may have influenced responses, although the experienced and trained multinational team of researchers at each site visit likely helped mitigate this limitation. Handwritten logbooks were retrospectively reviewed and may have been misinterpreted. Finally, there is a lack of available outcome data to supplement study findings.

## Conclusion

Private and PNFP hospitals in Uganda face cross-cutting challenges to delivering timely, affordable, and effective surgical care. Pervasive workforce shortages and workforce maldistribution, high medical and non-medical costs of seeking care and lack of financial risk protection, and poor coordination of care are challenges of unique importance in these sectors. In addition to financial resources and infrastructure development, SOA workforce scale-up is highlighted as an urgent need. Historically, little attention has been paid to understanding and improving the capacity of the private and PNFP sector which must be a critical component of ongoing capacity building efforts going forward. Critical policy and programmatic developments are essential to coordinate and strengthen Ugandan surgical capacity.

## Supporting information

**S1 Appendix. Semi-Structured interview tool.**
(PDF)

## Acknowledgments

We are grateful to our collaborating partners and the hospitals and staff that participated in this assessment, without whose help and enthusiasm this study would not have been possible. Many thanks in particular to our colleagues at the MOH and Ugandan Catholic Medical Bureau whose partnership ensured a comprehensive evaluation, access to facilities, and applicability and accessibility of results at the country level.

## Author Contributions

**Conceptualization:** Katherine Albutt, Didacus B. Namanya, Mark G. Shrime, Peter Kayima.

**Data curation:** Katherine Albutt, Gustaf Drevin, Didacus B. Namanya, Peter Kayima.

**Formal analysis:** Katherine Albutt, Gustaf Drevin, Rachel R. Yorlets, Emma Svensson.

**Funding acquisition:** Katherine Albutt.

**Investigation:** Didacus B. Namanya, Peter Kayima.

**Methodology:** Katherine Albutt, Didacus B. Namanya, Peter Kayima.

**Project administration:** Katherine Albutt, Gustaf Drevin, Mark G. Shrime, Peter Kayima.

**Supervision:** Katherine Albutt, Didacus B. Namanya, Mark G. Shrime.

**Validation:** Katherine Albutt, Rachel R. Yorlets, Emma Svensson, Peter Kayima.

**Writing – original draft:** Katherine Albutt, Gustaf Drevin, Rachel R. Yorlets, Emma Svensson.

**Writing – review & editing:** Katherine Albutt, Gustaf Drevin, Rachel R. Yorlets, Emma Svensson, Didacus B. Namanya, Mark G. Shrime, Peter Kayima.

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
