## [Decision Letter · Decision Letter 0]

16 Aug 2019

PONE-D-19-19864

‘We are all serving the same Ugandans’: a nationwide mixed-methods evaluation of private sector surgical capacity in Uganda

PLOS ONE

Dear Ms Albutt,

Thank you for submitting your manuscript to PLOS ONE. After careful consideration, we feel that it has merit but does not fully meet PLOS ONE’s publication criteria as it currently stands. Therefore, we invite you to submit a revised version of the manuscript that addresses the points raised during the review process.

We would appreciate receiving your revised manuscript by 30th September 2019. To enhance the reproducibility of your results, we recommend that if applicable you deposit your laboratory protocols in protocols.io, where a protocol can be assigned its own identifier (DOI) such that it can be cited independently in the future. For instructions see: http://journals.plos.org/plosone/s/submission-guidelines#loc-laboratory-protocols

We look forward to receiving your revised manuscript.

Kind regards,

Kwasi Torpey, MD PhD MPH

Academic Editor

PLOS ONE

Journal Requirements:

1. Please amend your current ethics statement to address the following concerns: Please explain why was written consent was not obtained, how you recorded/documented participant consent, and if the ethics committees/IRBs approved this consent procedure.

2. Please amend your list of authors on the manuscript to ensure that each author is linked to an affiliation. Authors’ affiliations should reflect the institution where the work was done (if authors moved subsequently, you can also list the new affiliation stating “current affiliation:….” as necessary).

Reviewers' comments:

Reviewer's Responses to Questions

**Comments to the Author**

1. Is the manuscript technically sound, and do the data support the conclusions?

Reviewer #1: Yes

Reviewer #2: No

2. Has the statistical analysis been performed appropriately and rigorously? 

Reviewer #1: Yes

Reviewer #2: No

3. Have the authors made all data underlying the findings in their manuscript fully available?

Reviewer #1: No

Reviewer #2: Yes

4. Is the manuscript presented in an intelligible fashion and written in standard English?

Reviewer #1: Yes

Reviewer #2: Yes

5. Review Comments to the Author

Reviewer #1: This paper provides a capacity assessment for surgical care at private Ugandan hospitals, supplemented by qualitative data from key informants to give greater context to the challenges facilities face in service delivery. It addresses a need to understand the private hospital system that is often neglected or very difficult to obtain information from in similar settings. The paper is clear and well-structured with only some points needing clarification, but could do more to move the conversation forward towards actionable solutions and/or outlining next steps on the agenda.

Major comments:

1. On sampling: Would be additionally helpful to understand the total distribution of public and private/PNFP hospitals across Uganda and understand what proportion of private facilities have been surveyed here. May also be relevant to have some understanding of who operates the private hospitals in the country – is it two or three major private entities or is facility operated by completely different entities? Understanding these aspects may lend greater credence to representativeness of the sample.

2. In the implementation of the quantitative tool, were each of the items/processes/individuals in question observed or were individuals interviewed to answer these questions? This is implied, but it would be helpful to directly state this is indeed the case.

3. Were group interviews managed or structured differently from individual interviews? For example, was it moderated in such a way as to ensure participation from all in the group? Or in other ways? What consideration was there of potential group effects (e.g. hierarchical structures among participants of a mixed group)? Or were group interviews conducted by cadre?

4. For qualitative interviews, it would be helpful to see a breakdown of the type of respondent by type of hospital (i.e. large/small, Private/PNFP, Bellwether/non-Bellwether facility, other relevant categories) being cautious not to allow for the identification of respondents. If only certain cadres responded for a particular type of facility, it may be a limitation of having only a single perspective for such cases.

5. Where are measures on ‘access to care’ as reported in the results collected across the quantitative and qualitative tools? It is unclear from where this information has been drawn given the tools’ focuses on infrastructure, service delivery, workforce, information management, and financing.

6. On supply chain, is there information on what proportion of supplies are obtained via Joint Medical Stores, donations, and the open market? This could point to how much donations are relied upon, for example, and what issues to address to improve the supply chain (e.g. funding, pricing, or other agreements with stores, etc.).

7. Across the paper, it would be helpful to distinguish when you are referring to clinical providers or the organization as provider, if that is the case. At some points, it is unclear whether answers came from clinical providers or hospital administrators (i.e, hospital directors, finance officers, etc.), though they would have distinctly different views and experiences on topics, such as staff training, contracts, service delivery, and costs.

8. Across the paper, it would be helpful in several cases to specify the numbers and/or percentages instead of stating “many hospitals” or “most hospitals”. This would provide clearer information more directly.

9. The barriers presented here are common in resource-constrained settings, and the authors have sufficiently linked their findings to other literature on the Ugandan health system. For at least some of these issues, the discussion could go on to discuss potential solutions that have been proposed or attempted in similar settings to move their agenda forward. There is some discussion comparing the public and private sectors – is there potential for solutions across public and private sectors given the larger understanding of the broader surgical care system that the information presented in the paper provides?

Minor comments:

1. Clarification requested on data collection, line 87: What was the sampling frame from which these private/PNFP hospitals were selected? Are we to assume that the MOH has a comprehensive list of all private hospitals in the country which was used to select the sample? This may not be the case in some resource-constrained settings.

2. Line 176: Should be a comma, not a period after ‘ward’.

3. To provide greater understanding of an individual’s perspective, could provide some description of the respondent’s cadre when providing quotes from qualitative interviews.

4. Line 241: It would be more helpful to identify how many facilities of those surveyed had set up this ‘stand-by’ group and expand on what proportion of providers in this group are split between public and private service delivery.

5. The results of t-tests run for comparisons are not presented or described in the text? For which comparisons were these statistical tests conducted? And what were the findings?

6. Figure 1: Could some indication of population density be included in the map?

Other general comments / questions:

1. Is the distribution of public and private/PNFP hospitals across Uganda proportional to care utilization patterns? For example, if about half of Ugandans access care in the private sector, do private facilities also make up half of the country’s facilities or is something else driving disproportionate utilization? If there are fewer private facilities, but half of utilization still happens at these facilities, it may provide yet greater motivation for understanding the capacity of these facilities to provide quality care with consideration of demand.

2. Were there differences in barriers or trends in capacity by region or type of hospital?

Reviewer #2: This article by Albutt, et al, provides a comprehensive assessment of capacity that includes evaluation in the context of bellwethers, operative log assessment, and structured interview. The overall goal is working in collaboration with the MOH to improve care, and the collaborative nature of the work is to be applauded. However, the manuscript lacks focus in the results section. It is unclear whether results are frim the quantitative or qualitative portion of the study and hence, reads more like a public health report than a scientific manuscript. I would recommend to the authors to completely re-write/re-present the results in an organized and systematic fashion with as much data presented as possible rather than summary results. This study has great value, but it needs some work in more deliberate presentation of results.

Some additional comments are listed below.

Abstract conclusion should read “As in Uganda’s public sector”, not “As Uganda’s public sector”

Line 73-74 run-on sentence

Line 75-77 reference that 29% of facilities in prior studies were PNFP but lists a huge number of references. Is this a meta-analysis or a result from one study? As much as I appreciate the extensive literature review of the authors, referencing 9-38 references is nonspecific.

Line 85 Is there a reference to the prior study? If not, then the methods need to b fully described.

Line 89 Is the stratified random sampling method utilized by the authors an established metholdology (i.e., provide a reference for the methodology).

Line 90-93 How were the hospitals randomly sampled? Please describe the sampling methodology in detail. Were they randomly sampled or a convenience sample using criteria?

Line 129 “Interview transcriptions were cross-checked for accuracy and completeness.” By whom? How were they checked?

While I found the study overall to be very comprehensive and collaborative, I found that the results were nondescript. The study uses qualitative and quantitiative methodology, and it was unclear which was used for the presentation of results.

It is unclear in the results what is taken from the quantitative survery and what is evaluated from the qualitiative assessment. For example, “often” under access—is this quantitative or qualititative? Both are described in the methodology. The catchment areas described, is this a known number (which it should be?) or an estimate form the qualitative assessment?

Line 171 (as an example) Please provide quantitative results in table format.

Line 182 “Most facilities” – please be more definitive. This “most” is repeated throughout the results rather than definitive numbers.

Despite the first portion of the study being a quantitative assessment of capacity, including resources, and procedures, the results section utilizes nondescript “most” rather than numerical results. It is unclear throughout the results section which results are qualitative and which are quantitative. As such, the vision of the background and methods is diluted.

Line 259-261 If C-sections are 48% and laparotomies and open fracture reduction are only 4.9% and 2/0%, respectively, what are the other procedures being performed? Are they “major” or “minor” surgery? Does this reflect anything. then, on the selection of these procedures as bellwethers?

Line 262 “This is particularly common for orthopedic procedures: patients are

263 often temporized and referred, asked to wait for an orthopedic specialist to be called in from

264 another hospital, or are treated by a non-specialist.” Again, it is unclear if this is reported in the qualititate portion fo the study or if this is a conclusion drawn from the study, in which case, it should be in the discussion.

266-268 Please quantify this better, ideally in table format.

The discussion is well-written and a good summary.

What does this study add to the literature? The discussion should focus more on how this study is different and how it adds to what is already known in the literature.

6. PLOS authors have the option to publish the peer review history of their article (what does this mean?). If published, this will include your full peer review and any attached files.

Reviewer #1: No

Reviewer #2: No

---

## [Author Response · Author response to Decision Letter 0]

2 Oct 2019

Please see attached response to reviewers

---

## [Editor Report · Decision Letter 1]

9 Oct 2019

‘We are all serving the same Ugandans’: a nationwide mixed-methods evaluation of private sector surgical capacity in Uganda

PONE-D-19-19864R1

Dear Dr. Albutt,

We are pleased to inform you that your manuscript has been judged scientifically suitable for publication and will be formally accepted for publication once it complies with all outstanding technical requirements.

With kind regards,

Kwasi Torpey, MD PhD MPH

Academic Editor

PLOS ONE
---

## [Editor Report · Acceptance letter]

17 Oct 2019

PONE-D-19-19864R1 

‘We are all serving the same Ugandans’: a nationwide mixed-methods evaluation of private sector surgical capacity in Uganda 

Dear Dr. Albutt:

I am pleased to inform you that your manuscript has been deemed suitable for publication in PLOS ONE. Congratulations! Your manuscript is now with our production department. 

With kind regards,

on behalf of

Professor Kwasi Torpey 

Academic Editor

PLOS ONE